

# Forecasting CO₂ emissions of fuel vehicles for an ecological world using ensemble learning, machine learning, and deep learning models

Fatih Gurcan

Department of Management Information Systems, Faculty of Economics and Administrative Sciences, Karadeniz Technical University, Trabzon, Turkey

## ABSTRACT

**Background**. The continuous increase in carbon dioxide ($CO_2$) emissions from fuel vehicles generates a greenhouse effect in the atmosphere, which has a negative impact on global warming and climate change and raises serious concerns about environmental sustainability. Therefore, research on estimating and reducing vehicle $CO_2$ emissions is crucial in promoting environmental sustainability and reducing greenhouse gas emissions in the atmosphere.

**Methods**. This study performed a comparative regression analysis using 18 different regression algorithms based on machine learning, ensemble learning, and deep learning paradigms to evaluate and predict $CO_2$ emissions from fuel vehicles. The performance of each algorithm was evaluated using metrics including $R^2$, Adjusted $R^2$, root mean square error (RMSE), and runtime.

**Results**. The findings revealed that ensemble learning methods have higher prediction accuracy and lower error rates. Ensemble learning algorithms that included Extreme Gradient Boosting (XGB), Random Forest, and Light Gradient-Boosting Machine (LGBM) demonstrated high $R^2$ and low RMSE values. As a result, these ensemble learning-based algorithms were discovered to be the most effective methods of predicting $CO_2$ emissions. Although deep learning models with complex structures, such as the convolutional neural network (CNN), deep neural network (DNN) and gated recurrent unit (GRU), achieved high $R^2$ values, it was discovered that they take longer to train and require more computational resources. The methodology and findings of our research provide a number of important implications for the different stakeholders striving for environmental sustainability and an ecological world.

# INTRODUCTION

In the present day, carbon dioxide ($CO_2$) emissions from fossil fuel-powered vehicles are a serious concern for environmental sustainability. The increasing number of vehicles on the roads and the growing need for mobility continuously increase the amount of greenhouse gases released into the atmosphere. In recent years, the global increase in

Corresponding author
Fatih Gurcan, fgurcan@ktu.edu.tr

population and economic growth has increased the demand for vehicles, leading to an increase in $CO_2$ emissions (*Yoro & Daramola, 2020*; *Kumar, Nagar & Anand, 2021*; *Hien & Kor, 2022*; *Zhang et al., 2022*). Especially in developing countries, the growing use of fuel vehicles, and the widespread use of gasoline and diesel vehicles in these countries, have led to an increasing amount of $CO_2$ being emitted into the atmosphere (*Suarez-Bertoa et al., 2020*; *Cha et al., 2021*; *Kumar, Nagar & Anand, 2021*). This worrying increase in $CO_2$ emissions brings various environmental problems such as global warming, melting glaciers, rising sea levels, climate change, and natural disasters (*Le Cornec et al., 2020*; *Yoro & Daramola, 2020*; *Kumar, Nagar & Anand, 2021*; *Zhong et al., 2021*; *Zhang et al., 2022*). Awareness of these environmental issues has led many countries and industries to take action. Globally, various regulations and standards are implemented to limit and reduce vehicle $CO_2$ emissions (*Ding et al., 2017*; *Yoro & Daramola, 2020*; *Kumar, Nagar & Anand, 2021*; *Hien & Kor, 2022*; *Tripp-Barba et al., 2023*). The automotive sector is taking significant steps towards investing in more environmentally friendly technologies and promoting zero-emission vehicles (*Suarez-Bertoa et al., 2020*; *Zhang & Fujimori, 2020*; *Zhang et al., 2022*; *Gurcan et al., 2023*).

The increasing environmental awareness has led academic research in the context of $CO_2$ emissions to focus more on this environmental issue (*Zhong et al., 2021*). Especially, the rapid advancements in software engineering, information systems, and artificial intelligence technologies have brought new approaches and perspectives to academic research in this field (*Zhong et al., 2021*; *Gurcan et al., 2022b*; *Gurcan, 2023*). In recent times, a large number of studies have concentrated on the use of machine learning and deep learning to forecast CO2 emissions (*Ding et al., 2017*; *Seo et al., 2021*; *Hien & Kor, 2022*; *Wang et al., 2022*; *Al-Nefaie & Aldhyani, 2023*; *Natarajan et al., 2023*). The approaches of machine learning, which enable the analysis of large amounts of data and the detection of complex relationships, have been used in many studies to more accurately predict vehicle $CO_2$ emissions (*Cha et al., 2021*; *Seo et al., 2021*; *Zhong et al., 2021*; *Madziel et al., 2022*; *Tansini, Pavlovic & Fontaras, 2022*; *Al-Nefaie & Aldhyani, 2023*; *Natarajan et al., 2023*). A number of machine learning algorithms have previously been implemented in a number of research studies to identify factors such as vehicle type, engine characteristics, fuel type, driving conditions, and their relationships to describe vehicle $CO_2$ emissions (*Ding et al., 2017*; *Le Cornec et al., 2020*; *Wang et al., 2022*; *Al-Nefaie & Aldhyani, 2023*; *Natarajan et al., 2023*; *Tena-Gago et al., 2023*).

*Cha et al. (2021)* conducted an experimental study based on regression analysis to predict real-world $CO_2$ emissions of light-duty diesel vehicles. *Madziel et al. (2022)* applied various machine learning models on data obtained from hybrid electric cars to predict $CO_2$ emissions and found that Gaussian process regression was the most effective approach. *Al-Nefaie & Aldhyani (2023)* used a deep learning model to predict $CO_2$ emissions from traffic vehicles in sustainable and smart environments. They achieved an $R^2$ value of 93.78 with their applied BiLSTM model. Using various machine learning algorithms, *Natarajan et al. (2023)* carried out an experimental regression analysis in a recent study to predict carbon dioxide emissions from light-duty vehicles. They made inferences about predicting $CO_2$ emissions with reasonable accuracy using the Catboost algorithm.

*Ding et al. (2017)* conducted a study to predict China's fuel combustion $CO_2$ emissions using a new multivariate model and compared their proposed model with four benchmark models. *Le Cornec et al. (2020)* conducted a study focused specifically on predicting NOx emissions, modeling and analyzing real-time emissions from diesel vehicles using machine learning. *Tansini, Pavlovic & Fontaras (2022)* conducted an analysis to measure real-world $CO_2$ emissions and energy consumption of plug-in hybrid vehicles. As a result, they proposed a three-dimensional $CO_2$ emission model based on the state of charge and average journey speed or wheel energy. *Seo et al. (2021)* used a methodology that combined an ANN model and a vehicle dynamics model to predict real-world carbon dioxide emissions from light-duty diesel vehicles. *Tena-Gago et al. (2023)* presented a new long short-term memory (LSTM)-based model called UWS-LSTM using traditional and advanced machine learning (ML) models to predict $CO_2$ emissions from hybrid vehicles. Using a portable emission measurement system (PEMS), researchers in another study examined the fuel consumption and exhaust emissions of a hybrid electric vehicle (HEV) in real-world settings (*Wang et al., 2022*). Using only the kinematic data of the vehicle, a gradient boosting model was created in this context to forecast the emissions and operation of the electric motor (*Wang et al., 2022*). Using a similar viewpoint, *Hien & Kor (2022)* developed a variety of regression models and convolutional neural networks (CNN) to forecast fuel consumption and carbon dioxide emissions for upcoming car production.

The estimation of vehicle $CO_2$ emissions using machine learning can provide valuable insights into reducing the impacts of such environmental issues (*Zhong et al., 2021*; *Hien & Kor, 2022*; *Tansini, Pavlovic & Fontaras, 2022*; *Gurcan, 2023*). More accurate emission predictions contribute to the development of environmental-friendly policies and strategies, leading to a reduction in greenhouse gas emissions (*Suarez-Bertoa et al., 2020*; *Yoro & Daramola, 2020*; *Kumar, Nagar & Anand, 2021*). Estimating $CO_2$ emissions with machine learning helps in developing strategies to increase vehicle energy efficiency and reduce fuel consumption (*Mardani et al., 2020*; *Zhang et al., 2022*). The accuracy of these predictions enhances the effectiveness of policies and strategies implemented across various sectors, facilitating steps towards a more sustainable future (*Chicco, Warrens & Jurman, 2021*; *Zhong et al., 2021*). While significant efforts have been made to improve the predictive performance of proposed models for estimating $CO_2$ emissions, there is still room to create alternative models in an interpretable manner, enhance prediction performance, and overcome limitations (*Cha et al., 2021*; *Seo et al., 2021*; *Zhong et al., 2021*; *Madziel et al., 2022*; *Tansini, Pavlovic & Fontaras, 2022*; *Al-Nefaie & Aldhyani, 2023*; *Natarajan et al., 2023*). The aim of these studies is to provide scientifically based policy recommendations to reduce vehicle $CO_2$ emissions and support the transition to a more sustainable transportation system (*Ding et al., 2017*; *Le Cornec et al., 2020*; *Le Cornec et al., 2020*; *Cha et al., 2021*; *Al-Nefaie & Aldhyani, 2023*; *Tena-Gago et al., 2023*). Such research is crucial for reducing environmental impacts, increasing energy efficiency, and combating climate change (*Suarez-Bertoa et al., 2020*; *Kumar, Nagar & Anand, 2021*; *Zhong et al., 2021*).

Considering this background, this article aims to address these shortcomings and aims to establish a robust model based on artificial intelligence technology capable of predicting $CO_2$ emissions from fuel vehicles. To achieve this goal, a wide range of supervised regression

algorithms and feature selection methods with different characteristics are included in the experimental analysis of this study. In this context, a large-scale forecasting analysis was conducted using 18 different regression algorithms from ensemble learning (EL), ML, and deep learning (DL) models. The performance metrics of each algorithm, including $R^2$, Adjusted $R^2$, root mean square error (RMSE), and runtime, were computed to evaluate the regression models. In conclusion, the research questions guiding this study can be summarized as follows:

RQ1. How effectively can different machine learning, ensemble learning, and deep learning algorithms predict the $CO_2$ emissions of fuel vehicles?

RQ2. Which supervised methods and learning paradigms provide the highest prediction accuracy and lowest error rates while balancing computational complexity, training time, and overall performance?

The remainder of this article is organized as follows. The "Materials and Methods" section describes the methodology of the study in detail. The "Experimental Results and Discussions" section presents the experimental results of the study and provides a comprehensive discussion. Finally, "Conclusions" summarizes key findings and future work.

## MATERIALS AND METHODS

### Data collection, preparation, and description

The dataset utilized in this analysis came from the "Fuel consumption rating" databases continued by the Canadian government. These databases contain assessments of fuel consumption and measured $CO_2$ emissions for a sample of 7,385 fuel-powered light commercial vehicles in Canada. This dataset is available *via* both the Canadian government's official link (*Government of Canada, 2024*) and the Kaggle open data source (*Podder, 2020*). The data set includes measurements for seven years, from 2014 to 2020, totaling 12 features for 7,384 fuel vehicle records. The dataset's initial form has 12 columns, known as attributes, which include five object columns, three integer columns, and four float columns. The dataset contains data for 2,053 different car models from 42 different automobile brands. Variables or attributes were reorganized and transformed into a more meaningful structure before being analyzed. First, missing values in the dataset were checked, and no missing values were discovered. Since there was only one vehicle using natural gas fuel in the data set, this record was eliminated. The dataset was checked for duplicate data, revealing 6,281 unique and 1,103 duplicate records. Subsequently, categorical variables of object type were restructured, and sub-categorization of these variables was performed. More precisely, the Transmission variable is divided into five subcategories: "Transmission_Manual", "Transmission_CVT", "Transmission_Automatic_Selective", "Transmission_Automatic", and "Transmission_Automated_Manual". The Fuel Type variable is divided into four subcategories: "Fuel_Type_Diesel", "Fuel_Type_Ethanol", "Fuel_Type_Premium_Gasoline", and "Fuel_Type_Regular_Gasoline". Make types from 42 different automobile brands are divided into five subcategories: "Make_Type_Brand_High_Performance",

**Table 1** Variables and types in the dataset.

| ID | Variable | Data Type |
|---|---|---|
| X1 | Engine_Size | float64 |
| X2 | Cylinders | float64 |
| X3 | Fuel_Consumption_City | float64 |
| X4 | Fuel_Consumption_Hwy | float64 |
| X5 | Fuel_Consumption_Comb | float64 |
| X6 | Fuel_Consumption_Comb_MPG | float64 |
| X7 | Fuel_Type_Diesel | uint8 |
| X8 | Fuel_Type_Ethanol | uint8 |
| X9 | Fuel_Type_Premium_Gasoline | uint8 |
| X10 | Fuel_Type_Regular_Gasoline | uint8 |
| X11 | Transmission_Automated Manual | uint8 |
| X12 | Transmission_Automatic | uint8 |
| X13 | Transmission_Automatic_Selective | uint8 |
| X14 | Transmission_CVT | uint8 |
| X15 | Transmission_Manual | uint8 |
| X16 | Make_Type_Brand_High_Performance | uint8 |
| X17 | Make_Type_Brand_Luxury | uint8 |
| X18 | Make_Type_Brand_Mainstream | uint8 |
| X19 | Make_Type_Brand_Premium | uint8 |
| X20 | Make_Type_Brand_Premium_Plus | uint8 |
| X21 | Vehicle_Class_Type_Compact | uint8 |
| X22 | Vehicle_Class_Type_SUV | uint8 |
| X23 | Vehicle_Class_Type_Sedan | uint8 |
| X24 | Vehicle_Class_Type_Truck | uint8 |
| Y | CO2_Emissions | float64 |

"Make_Type_Brand_Luxury", "Make_Type_Brand_Mainstream", "Make_Type_Brand_Premium", and "Make_Type_Brand_Premium_Plus". As a final point, 16 different vehicle classes were categorized into four subcategories: "Vehicle_Class_Type_Truck", "Vehicle_Class_Type_Sedan", "Vehicle_Class_Type_SUV", and "Vehicle_Class_Type_Compact". As a result of the dataset improvements, the total number of variables increased from 12 to 25. One of these variables represents dependent variables, while the other 24 represent independent variables. Table 1 displays the IDs, variable names, and data types for these variables.

## Data preprocessing and feature engineering

Data preprocessing involves the cleaning, transforming, and preparing of the dataset before providing it to a regression model (*Gurcan et al., 2022a*; *Al-Nefaie & Aldhyani, 2023*; *Gurcan, 2023*). It is an essential process for determining the variables in the regression model and enhancing the accuracy and performance of regression analysis. Categorical variables cannot be used directly in regression models. Therefore, categorical variables must be converted into numerical data. In this context, categorical variables of the object type have been transformed (*Raschka & Mirjalili, 2017*). The process of transformation may include

one-hot encoding, label encoding, or the embedded representation of categorical variables (*Scikit-learn, 2024*). For this dataset, the encoding model "One Hot Encoding", which represents categorical variables as binary, was used. Many regression analysis algorithms are sensitive to scale differences between features, so feature scaling or normalization is important (*Raschka & Mirjalili, 2017*; *Géron, 2019*). As a result, the experimental dataset was processed using the Min-Max feature scaling technique.

Following that, the feature selection stage was initiated. Feature selection in regression analysis reduces model complexity, improves generalizability, and increases performance (*Chicco, Warrens & Jurman, 2021*; *Al-Nefaie & Aldhyani, 2023*; *Natarajan et al., 2023*). During regression analysis, feature selection ensures that the dependent and independent variables are correctly identified. The dependent variable is the one you want to predict or explain, and the independent variables are those that describe it and have the potential to influence it. In this study, the dependent variable is $CO_2$ emissions. There are several feature selection techniques available for identifying independent variables (*Scikit-learn, 2024*).

In the present analysis, the feature selection process started with a correlation analysis. Correlation analysis was used to determine pairwise relationships between all variables. The dataset was then subjected to feature selection methods such as Chi-square to recognize the impact of independent variables on the dependent variable. This approach made it easier to identify the independent variables that had the greatest impact on the regression models that would be constructed (*Géron, 2019*; *Scikit-learn, 2024*). In addition, backward elimination and forward selection methods were used to determine the variables that best describe the model (*Raschka & Mirjalili, 2017*; *Scikit-learn, 2024*). As a result of the methods implemented during the feature selection phase, five variables with the least impact ("Transmission_Automated Manual", "Fuel_Type_Diesel", "Make_Type_Brand_Premium_Plus", "Vehicle_Class_Type_Sedan", and "Make_Type_Brand_Premium") were removed from the independent variables. The aforementioned approach of feature selection reduces the model's complexity, shortens training time, improves model generalization, and prevents overfitting (*Hien & Kor, 2022*; *Natarajan et al., 2023*). Following the feature selection phase, 19 independent variables that best described the regression model and its dependent variable were identified.

The dataset was then split into training and testing sets. The dataset for machine learning-based regression models is divided into two parts: training and testing. The training set trains the model, and the testing set evaluates its performance. This splitting process is critical for evaluating the model's generalization ability. As a result, for this analysis, 80% of the data was used for training and 20% for testing (*Scikit-learn, 2024*).

## Regression model adaptation and implementation

A machine learning-based multivariate regression model is a technique for analyzing the impact of one or more independent variables on the dependent variable. A multivariate regression model is a fundamental statistical method and a component of machine learning (*Le Cornec et al., 2020*; *Natarajan et al., 2023*). Unsupervised learning algorithms use unlabeled data to identify patterns, relationships, or groups (*Gurcan et al., 2022a*; *Gurcan,*

*Dalveren & Derawi, 2022*). Clustering, dimensionality reduction, semantic mapping, and topic modeling are some of the techniques used (*Zhong et al., 2021*; *Gurcan et al., 2023*). Regression algorithms use labeled data to create a model that associates an output variable with one or more input variables. These models predict the numerical value of a specific output variable. Regression models are classified as supervised learning because they use labeled data (*Huang et al., 2020*; *Uras et al., 2020*). This means that the model can produce more accurate and consistent results (*Le Cornec et al., 2020*; *Mardani et al., 2020*; *Cha et al., 2021*).

A multivariate regression model is used to explain the effect of several independent variables on the dependent variable. As an instance, it can be used to predict the price of a house by using various house characteristics such as size, age, location, and number of rooms as independent variables and the house's price as the dependent variable (*Chicco, Warrens & Jurman, 2021*; *Scikit-learn, 2024*). Machine learning-based multiple regression models, unlike traditional regression techniques, can handle larger and more complex datasets. Furthermore, by leveraging the various benefits of machine learning algorithms, they can develop more flexible and generalizable models (*Raschka & Mirjalili, 2017*; *Géron, 2019*).

This experimental analysis uses machine learning-based multivariate regression models to predict a vehicle's $CO_2$ emissions (g/km) based on fuel consumption and other descriptive features. In this case, a multivariate linear regression (MLR) model based on the ordinary least squares (OLS) method was applied to the experimental dataset to determine the effect of each independent variable on $CO_2$ emissions (*Statsmodels, 2024*). Each variable's significance for $CO_2$ emissions has been evaluated using this method.

In order to perform a comparative analysis of regression algorithms with various learning models, the dataset was subsequently subjected to regression models established with 18 different algorithms (three linear and 15 nonlinear) (*Raschka & Mirjalili, 2017*; *Nelli, 2023*; *Scikit-learn, 2024*). Six of these algorithms rely on ensemble learning, six on traditional machine learning, and six on deep learning techniques (*Tramontana et al., 2016*; *Raschka & Mirjalili, 2017*; *Géron, 2019*; *Huang et al., 2020*; *Scikit-learn, 2024*). The present investigation employed Python-based data science and machine learning libraries, which include Numpy, Seaborn, Matplotlib, Scikit-Learn, Statsmodel, Keras, TensorFlow, and Pandas, to execute data preprocessing, feature engineering, and regression analysis procedures. For the implementation of regression models and their hyper parameter settings, the documentation of Scikit-Learn, TensorFlow, and Keras libraries can be used. Additionally, the source codes we provide as supplementary files can serve as a guide for researchers. (*Raschka & Mirjalili, 2017*; *Géron, 2019*; *Nelli, 2023*; *Scikit-learn, 2024*; *Statsmodels, 2024*). We now describe the 18 regression models that were implemented to this analysis in three subsections.

## Ensemble learning-based regression models

Ensemble learning is a machine learning technique that combines multiple learning algorithms to produce a more robust and generalized model (*Madziel et al., 2022*; *Wang et al., 2022*; *Natarajan et al., 2023*). The fundamental principle of ensemble learning is based

on the idea that a collection of weak models can combine to form a stronger model. Each weak model learns a specific feature or subset of the data. The predictions of these weak models are then combined to create a more robust and generalized prediction (*Raschka & Mirjalili, 2017*; *Géron, 2019*; *Scikit-learn, 2024*).

The six ensemble learning algorithms employed in this analysis are described as follows:

- **XGB:** This ensemble learning model uses the Extreme Gradient Boosting (XGBoost) algorithm to perform regression analysis. XGBoost is based on the gradient boosting method and is well-known for its high performance, scalability, and generalization capabilities. XGBoost uses multiple decision trees to build a powerful predictive model. Each tree is trained to correct the errors of previous trees, resulting in increasingly accurate predictions. As a result, XGB typically outperforms other regression models in terms of prediction accuracy.

- **RandomForest:** This is a regression model used in the field of machine learning. This model makes use of the ensemble learning technique, which builds a potent predictor by combining several decision trees. RandomForest is an ensemble model consisting of a series of decision trees, each trained on a different subset. Regression problems are solved by each tree by teaching it the relationship between independent and dependent variables.

- **LGBM:** This is a regression model used as part of the LightGBM library. The abbreviation "LGBM" stands for "Light Gradient Boosting Machine". LightGBM is an open-source gradient boosting framework developed by Microsoft, designed to provide fast and high-performance machine learning on large-scale datasets. An ensemble learning method called gradient boosting has been employed in this model. With gradient boosting, a weak predictor (usually a decision tree) is combined to produce a strong predictor.

- **HistGradientBoosting:** This is an ensemble learning-based regression model, abbreviated as "Histogram-Based Gradient Boosting". This model applies the gradient boosting method using a histogram-based approach. HistGradientBoosting is designed to provide high speed and low memory usage when working with large-scale datasets. In this model, histograms of the dataset's features are used to construct decision trees. This captures the distribution of the dataset more effectively and allows for faster training times.

- **GradientBoosting:** It makes use of the gradient boosting method, an ensemble learning strategy that builds strong predictors by combining weak learners, usually decision trees. The fundamental principle of GradientBoosting involves sequentially training a series of weak predictors (such as decision trees), with each predictor trained to correct the errors of its predecessor. The GradientBoosting uses decision trees by default.

- **AdaBoost:** It stands for "Adaptive Boosting" in short. AdaBoost is an ensemble learning method that builds a strong predictor by combining weak learners, typically decision trees. The adaptive boosting (AdaBoost) algorithm, which entails training weak learners successively, is implemented by the model. Each learner is trained to correct the errors of its predecessor. AdaBoost defaults to using decision trees as weak learners, although different base estimator options are available.

## Deep learning-based regression models

Deep learning can be a powerful tool for modeling complex relationships and achieving high accuracy in regression problems. Depending on the complexity of the regression problem and the characteristics of the dataset, an appropriate deep learning model is chosen. This may include models such as multi-layer perceptrons (MLP), deep neural networks (DNN), or CNN (*Seo et al., 2021*; *Hien & Kor, 2022*; *Tena-Gago et al., 2023*). The selected model is then trained on the training dataset. Weights are updated using the backpropagation algorithm, improving the model's performance. This step typically requires large datasets and high computational power (*Le Cornec et al., 2020*; *Al-Nefaie & Aldhyani, 2023*). Therefore, the complexity of the model and the cost of the training process should be considered (*Raschka & Mirjalili, 2017*; *Géron, 2019*; *Scikit-learn, 2024*). The six deep learning algorithms utilized in this study are detailed below:

- **CNN:** This regression model uses convolutional neural networks architecture, commonly used for image data. CNNs are commonly utilized in the fields of image detection and processing. However, with proper feature engineering, they can also be used on numerical data. CNN outperforms classic regression models in handling complicated and high-dimensional data.

- **DNN:** It is a regression model based on deep neural networks. DNNs are a type of artificial neural network that has more than one hidden layer between its input and output. Because of their depth and nonlinear changes applied at each layer, these networks can capture intricate patterns and relationships within the data. DNN is especially useful for jobs that need big and high-dimensional datasets, where standard regression models may fail to deliver similar results.

- **GRU:** GRU is a regression model built with gated recurrent units (GRUs), a sort of recurrent neural network architecture. GRUs are intended to handle sequential and time-series data by retaining a hidden state that stores information from prior time steps. GRUs, unlike typical RNNs, have gating mechanisms that assist alleviate the vanishing gradient problem, making them better suited for learning long-term dependencies in data.

- **RNN:** RNN is a regression model that employs recurrent neural networks, a form of neural network architecture designed to handle sequential data. RNNs process sequences by keeping a hidden state that is updated at every time step, allowing the network to detect temporal dependencies and patterns in the input. This makes RNN ideal for regression problems involving time-series data, where the order and timing of observations are critical.

- **LSTM:** This regression model employs long short-term memory (LSTM) networks, a type of RNN designed to handle long-term dependencies in sequential data. The unique architecture of LSTM networks, which includes memory cells, input gates, output gates, and forget gates, addresses issues like the vanishing gradient problem that commonly plagues RNNs. LSTM networks can preserve and update memories over lengthy sequences, making them ideal for time-series regression tasks with temporal correlations.

- **MLP:** It is a deep learning-based model utilized to solve regression problems using multilayer perceptrons (MLP). This model generally performs well on datasets with complex and nonlinear relationships. MLP is the regression version of artificial neural networks and predicts output values based on input data. The training of the MLP model is performed using the backpropagation algorithm. This algorithm starts with the error at the model's output and then utilizes the chain rule of derivatives to update the weights inside the network backward.

## Machine learning-based regression models

Machine learning offers many methods and algorithms with different backgrounds to solve regression problems (*Mardani et al., 2020*; *Cha et al., 2021*; *Hien & Kor, 2022*). Conducting regression analysis using machine learning involves a series of steps, including selecting the appropriate algorithm and parameters, preparing the dataset, and training the model (*Tramontana et al., 2016*; *Raschka & Mirjalili, 2017*; *Géron, 2019*; *Scikit-learn, 2024*). In this experimental analysis, a total of six different algorithms based on both linear and non-linear models of traditional machine learning were utilized (*Scikit-learn, 2024*), and they are listed as follows:

- **DecisionTree:** It is a decision tree-based regression model used to solve regression problems. Decision trees partition the dataset based on simple decision rules over feature values to make regression predictions. The DecisionTree model applies decision rules at the nodes of the tree based on the values of features in the dataset, thereby making predictions for the dependent variable.
- **Ridge:** Ridge regression is based on the least squares method, but it adds a regularization term to reduce overfitting. This regularization term limits the magnitude of the model coefficients, thereby controlling overfitting. In Ridge regression, a lambda ($\lambda$) parameter is used to control overfitting.
- **LinearRegression:** This model expresses the relationship between the dependent variable and independent variables through a linear equation. The model calculates the predicted value of the dependent variable as a linear combination of the independent variables. The LinearRegression model estimates model parameters (coefficients) using the method of least squares. It finds the most suitable parameters by minimizing the mean squared error between the actual and predicted values.
- **KNeighbors:** A regression model for K-nearest neighbors regression. This model uses the average of neighboring points to predict the value of an instance. KNeighbors is based on the KNN (K-nearest neighbors) algorithm, which calculates an instance's predicted value using the nearest k neighbors. The nearest neighbors are usually determined using Euclidean distance or another similarity metric.
- **Lasso:** Lasso stands for Least Absolute Shrinkage and Selection Operator. Lasso regression is a linear modeling method used for regression analysis. Lasso is a generalization of Ridge regression (Tikhonov regularization). Lasso uses L1 regularization to control the complexity of the model. L1 regularization is employed to shrink the model coefficients towards zero, thereby eliminating unnecessary features and simplifying the model. This allows the model to be more interpretable and generalizable.

- **SVR:** Support vector regression (SVR) is a regression method that uses support vector machines (SVM) principles and is especially useful for modeling non-linear relationships. Its primary goal is to fit a region around the data points within a hyperplane, allowing the regression model to select the hyperplane that is closest to the data. Typically, this hyperplane includes some of the data points while increasing the margin.

## Performance evaluation of regression models

A wide range of metrics are commonly used to assess the performance of regression models. These metrics assess the model's ability to predict actual values and its overall success from various perspectives. However, because each problem and dataset is unique, careful analysis is required to choose the most appropriate performance metrics and evaluate the model. Performance evaluation assesses the model's predictive power (*Le Cornec et al., 2020*; *Hien & Kor, 2022*). As a result, evaluating regression model performance is an important step in data analysis and model development. The runtime, RMSE, $R^2$, and Adjusted $R^2$ metrics were used in this study to assess the performance of the regression models (*Chicco, Warrens & Jurman, 2021*; *Al-Nefaie & Aldhyani, 2023*; *Natarajan et al., 2023*). The runtime is the time it takes to apply each regression model to the experimental dataset. The RMSE metric is commonly used to assess the performance of regression models. RMSE is defined as the square root of the average of the squared differences between actual values and model predictions. Equation (1) provides a mathematical expression for RMSE (*Scikit-learn, 2024*). Here, $n$ represents the number of samples in the dataset, $y_j$ denotes the actual target values, $\hat{y}_j$ represents the model's predictions. RMSE involves computing the square root of the average of the squared differences between the predicted and actual values, which gives more weight to larger differences or errors, making RMSE more sensitive to outliers.

$$RMSE = \sqrt{\frac{1}{n}\sum_{j=1}^{n}(y_j - \hat{y}_j)^2}. \tag{1}$$

$R^2$ is a statistical measure used to evaluate how well a regression model fits the actual dataset. $R^2$ measures how well independent variables explain the dependent variable. $R^2$ values range from 0 to 1, with higher values indicating that the model better explains the dataset's variability. Equation (2) provides the mathematical representation of $R^2$ (*Scikit-learn, 2024*). Here, $n$ represents the number of samples in the dataset, $y_j$ represents the actual target values, $\hat{y}_j$ represents the model's predictions, and $\overline{y}_j$ represents the mean of the actual target values. *SSr* is the sum of squared residuals (*i.e.*, sum of squared errors), and *SSt* is the total sum of squares (*i.e.*, sum of squared deviations from the mean).

$$R^2 = 1 - \frac{SS_r}{SS_t} = 1 - \left(\frac{\sum_{j=1}^{n}(y_j - \hat{y}_j)^2}{\sum_{j=1}^{n}(y_j - \overline{y}_j)^2}\right). \tag{2}$$

Adjusted $R^2$ is a modified version of $R^2$ and provides a more accurate performance measurement by considering the complexity of regression models and overfitting tendencies. $R^2$ typically increases with the addition of each independent variable in the model, which may lead to overfitting. Adjusted $R^2$ addresses this issue and provides a more

robust performance measurement by considering the complexity of the model. Equation (3) is used to calculate Adjusted $R^2$ (*Scikit-learn, 2024*). In this case, $n$ represents the number of samples in the dataset, and $k$ represents the number of independent variables in the model. Adjusted $R^2$ may have a lower value than standard $R^2$, indicating that it considers the model's complexity and minimizes overfitting.

$$Adjusted R^2 = 1 - \left[ \left( \frac{n-1}{n-k-1} \right) \left( 1 - R^2 \right) \right]. \tag{3}$$

In the present study, $R^2$ and Adjusted $R^2$ were utilized concurrently to balance the overall accuracy and explanatory nature of the model with its complexity and generalization ability. From this perspective, the $R^2$ score was used to evaluate the overall performance, and the Adjusted $R^2$ score was used to minimize the risk of overfitting and to better understand the model's generalization capabilities. This approach allows for identifying not only the models with the highest $R^2$ scores but also the most balanced and generalizable models. In this analysis, both metrics were calculated, but the $R^2$ score was primarily used to evaluate the performance of the regression models.

# EXPERIMENTAL RESULTS AND DISCUSSIONS

## Correlation analysis and feature selection

In this section, we present the results of our analysis, beginning with the results of our correlation analysis to better understand the relationship between variables and how this relationship can be used for analysis or prediction. Correlation analysis is a statistical technique for determining the relationship between two variables. This analysis aids in determining the direction (positive, negative, or none), strength, and type (linear or non-linear) of the relationship between variables. Table 2 displays the results of our correlation analysis, which reveal the direction and strength of the relationship between the 24 independent variables and the dependent variable. The variables in the table are listed in descending order of correlation score. According to the table, the first six variables with numerical values have a high impact on $CO_2$ emissions. While the variable "Fuel_Consumption_Comb_MPG" has a strong inverse relationship with the dependent variable, the other top five variables show a strong and positive relationship in the same direction. On the other hand, categorical variables have been shown to have a much smaller effect on the dependent variable. Table A1, provided as a supplementary file, provides a correlation matrix table showing the relationships between all variables. Using the outcomes from this table, it's possible to analyze the strength and direction of the relationships between all variables.

In continuation of the analysis, the dataset was subjected to Chi-square ($CHI^2$), F-regression (FR), and mutual-information-regression (MIR) tests to determine the importance of independent variables for the target variable and to select the most effective features. The results from each test model were normalized using min-max scaling to ensure that each test was normalized. The average of the scores obtained from three different methods was then computed to produce a score for each independent variable, which is shown in Table 3. This method allowed us to present a novel approach to feature

**Table 2** Correlation between independent variables and target variable.

| ID | Variable | Correlation Score |
|---|---|---|
| X3 | Fuel_Consumption_City | 0.920 |
| X5 | Fuel_Consumption_Comb | 0.918 |
| X6 | Fuel_Consumption_Comb_MPG | −0.908 |
| X4 | Fuel_Consumption_Hwy | 0.884 |
| X1 | Engine_Size | 0.851 |
| X2 | Cylinders | 0.833 |
| X14 | Transmission_CVT | −0.329 |
| X24 | Vehicle_Class_Type_Truck | 0.310 |
| X12 | Transmission_Automatic | 0.267 |
| X18 | Make_Type_Brand_Mainstream | −0.264 |
| X10 | Fuel_Type_Regular_Gasoline | −0.260 |
| X16 | Make_Type_Brand_High_Performance | 0.234 |
| X9 | Fuel_Type_Premium_Gasoline | 0.231 |
| X21 | Vehicle_Class_Type_Compact | −0.219 |
| X15 | Transmission_Manual | −0.166 |
| X17 | Make_Type_Brand_Luxury | 0.150 |
| X22 | Vehicle_Class_Type_SUV | 0.121 |
| X23 | Vehicle_Class_Type_Sedan | −0.120 |
| X19 | Make_Type_Brand_Premium | 0.118 |
| X8 | Fuel_Type_Ethanol | 0.096 |
| X20 | Make_Type_Brand_Premium_Plus | 0.074 |
| X13 | Transmission_Automatic_Selective | 0.071 |
| X7 | Fuel_Type_Diesel | −0.035 |
| X11 | Transmission_Automated Manual | −0.005 |
| Y | CO2_Emissions | 1.00 |

evaluation and selection. As can be seen from the table, the variables related to fuel consumption rank in the top four and are therefore the most effective. These are followed by the variables "Engine_Size" and "Cylinders". Less important categorical variables are located at the bottom of the table and provide important clues about which variables will be eliminated first from the regression model.

## Linear regression using OLS

On this dataset, a multivariate linear regression model was implemented using Ordinary Least Squares (OLS). The model's outputs are summarized in Table 4. Part 1 of the output in Table 4 begins with a list of common model indicators. Df Residuals displays the degrees of freedom in our model, whereas Df Model shows the number of independent variables. $R^2$ is the most noteworthy metric generated by this regression model. $R^2$ is the proportion of the variance of the dependent variable that the independent variables can account for. The fact that our model explains 99.3% of the variation in the 'CO2_Emissions' variable indicates a high level of explanatory power.

In a linear regression model, the F-statistic is used to determine whether a set of variables is statistically significant in explaining the variance in the dependent variable within the

**Table 3  Scores of feature selection tests.**

| ID | Variable | CHI$^2$ | FR | MIR | Mean |
|---|---|---|---|---|---|
| X6 | Fuel_Consumption_Comb_MPG | 1.000 | 0.851 | 0.774 | 0.875 |
| X5 | Fuel_Consumption_Comb | 0.344 | 0.978 | 1.000 | 0.774 |
| X3 | Fuel_Consumption_City | 0.452 | 1.000 | 0.660 | 0.704 |
| X4 | Fuel_Consumption_Hwy | 0.222 | 0.649 | 0.512 | 0.461 |
| X1 | Engine_Size | 0.229 | 0.479 | 0.315 | 0.341 |
| X2 | Cylinders | 0.231 | 0.412 | 0.194 | 0.279 |
| X16 | Make_Type_Brand_High_Performance | 0.185 | 0.011 | 0.004 | 0.067 |
| X14 | Transmission_CVT | 0.115 | 0.022 | 0.020 | 0.053 |
| X24 | Vehicle_Class_Type_Truck | 0.081 | 0.019 | 0.022 | 0.041 |
| X17 | Make_Type_Brand_Luxury | 0.074 | 0.004 | 0.007 | 0.028 |
| X12 | Transmission_Automatic | 0.044 | 0.014 | 0.021 | 0.026 |
| X18 | Make_Type_Brand_Mainstream | 0.025 | 0.014 | 0.022 | 0.020 |
| X9 | Fuel_Type_Premium_Gasoline | 0.023 | 0.010 | 0.019 | 0.018 |
| X10 | Fuel_Type_Regular_Gasoline | 0.017 | 0.013 | 0.023 | 0.018 |
| X20 | Make_Type_Brand_Premium_Plus | 0.034 | 0.001 | 0.010 | 0.015 |
| X11 | Transmission_Automated Manual | 0.038 | 0.000 | 0.005 | 0.014 |
| X8 | Fuel_Type_Ethanol | 0.028 | 0.002 | 0.008 | 0.013 |
| X15 | Transmission_Manual | 0.024 | 0.005 | 0.011 | 0.013 |
| X21 | Vehicle_Class_Type_Compact | 0.020 | 0.009 | 0.010 | 0.013 |
| X22 | Vehicle_Class_Type_SUV | 0.020 | 0.003 | 0.012 | 0.012 |
| X19 | Make_Type_Brand_Premium | 0.010 | 0.003 | 0.018 | 0.010 |
| X23 | Vehicle_Class_Type_Sedan | 0.013 | 0.003 | 0.009 | 0.008 |
| X13 | Transmission_Automatic_Selective | 0.003 | 0.001 | 0.012 | 0.005 |
| X7 | Fuel_Type_Diesel | 0.000 | 0.000 | 0.000 | 0.000 |

linear model. The calculated F-value for this model (4.404e+04) is significantly high, indicating that the model is statistically significant for these variables. The likelihood that the provided data will be generated by the regression model is represented by the log-likelihood. In the process of creating a model, it is employed to compare each variable's coefficient values. When comparing the efficacy of models in linear regression, AIC and BIC are both employed to penalize the models using a penalty system for multiple variables. Omnibus defines the normality of our residuals' distribution as a measure using skewness and kurtosis. A value of 0 indicates perfect normality. An arithmetic test called Prob(Omnibus) determines the likelihood that residuals will have a normal distribution. A value of 1 indicates complete normality. In our data, skewness is a metric for symmetry, where 0 denotes perfect symmetry. Kurtosis quantifies the data's peak or characterizes the density surrounding a normal curve's zero point. There are fewer outliers in the data when the kurtosis is higher.

Durbin-Watson is a measure of homoscedasticity or the equal distribution of errors in the data. Varying variance indicates an unequal distribution; for example, as data points increase, relative errors also increase. An ideal measure for homoscedasticity is expected to be between 1 and 2. Alternative techniques known as Jarque–Bera (JB) and Prob(JB)

**Table 4  Results of OLS-based regression analysis.**

**OLS regression results (part 1)**

| Depended variable: | CO2_Emissions | R-squared: | 0.993 | Omnibus: | 1790.639 |
|---|---|---|---|---|---|
| Model: | OLS | Adj. R-squared: | 0.993 | Prob(Omnibus): | 0.000 |
| Method: | Least squares | F-statistic: | 4.404e+04 | Skew: | −0.597 |
| No. observations: | 5907 | Prob (F-statistic): | 0.00 | Kurtosis: | 24.296 |
| Df residuals: | 5887 | Log-Likelihood: | −17769. | Durbin-Watson: | 2.063 |
| Df model: | 19 | AIC: | 3.558e+04 | Jarque–Bera (JB): | 111974.406 |
| | | BIC: | 3.571e+04 | Prob(JB): | 0.000 |
| | | | | Cond. No. | 990. |

**OLS regression results (part 2)**

| Variables | Coef | Std Err | t | $P > |t|$ | [0.025 | 0.975] |
|---|---|---|---|---|---|---|
| Intercept | 85.9680 | 1.623 | 52.984 | 0.000 | 82.787 | 89.149 |
| Engine_Size | 0.3979 | 0.151 | 2.631 | 0.009 | 0.101 | 0.694 |
| Cylinders | 1.0262 | 0.111 | 9.228 | 0.000 | 0.808 | 1.244 |
| Fuel_Consumption_City | 5.8527 | 0.828 | 7.071 | 0.000 | 4.230 | 7.475 |
| Fuel_Consumption_Hwy | 4.4844 | 0.687 | 6.528 | 0.000 | 3.138 | 5.831 |
| Fuel_Consumption_Comb | 9.3286 | 1.501 | 6.214 | 0.000 | 6.385 | 12.272 |
| Fuel_Consumption_Comb_MPG | −0.8741 | 0.028 | −31.207 | 0.000 | −0.929 | −0.819 |
| Fuel_Type_Ethanol | −1367190 | 0.643 | −212553 | 0.000 | −137.980 | −135458 |
| Fuel_Type_Premium_Gasoline | −29.6778 | 0.486 | −61.106 | 0.000 | −30.630 | −28.726 |
| Fuel_Type_Regular_Gasoline | −29.8430 | 0.462 | −64.533 | 0.000 | −30.750 | −28.936 |
| Transmission_Automatic | −1.0174 | 0.289 | −3.524 | 0.000 | −1.583 | −0.451 |
| Transmission_Automatic_Selective | −0.7959 | 0.254 | −3.132 | 0.002 | −1.294 | −0.298 |
| Transmission_CVT | −1.0391 | 0.351 | −2.960 | 0.003 | −1.727 | −0.351 |
| Transmission_Manual | −1.0790 | 0.282 | −3.830 | 0.000 | −1.631 | −0.527 |
| Make_Type_Brand_High_Performance | 3.9416 | 0.636 | 6.199 | 0.000 | 2.695 | 5.188 |
| Make_Type_Brand_Luxury | 1.7326 | 0.351 | 4.941 | 0.000 | 1.045 | 2.420 |
| Make_Type_Brand_Mainstream | 0.6721 | 0.177 | 3.788 | 0.000 | 0.324 | 1.020 |
| Vehicle_Class_Type_Compact | −0.8753 | 0.172 | −5.083 | 0.000 | −1.213 | −0.538 |
| Vehicle_Class_Type_SUV | 0.6258 | 0.212 | 2.955 | 0.003 | 0.211 | 1.041 |
| Vehicle_Class_Type_Truck | 1.3381 | 0.321 | 4.164 | 0.000 | 0.708 | 1.968 |

measurements use skewness and kurtosis to measure the same value as Omnibus and Prob(Omnibus). These values are used to cross-validate the measurements. Condition number measures the sensitivity of the output of a regression model compared to the input. Among multicollinearity indicators, a high condition number value is the most important. The term "multicollinearity" refers to the presence of two or more independent variables that have a strong correlation with one another and may cause redundancy, which could inaccurately affect our predicted variable. In the case of multicollinearity, we can expect much higher fluctuations in the data with small changes. When all variables are included in the regression model without feature selection, the Cond_No value was calculated as 1.84e+17. Following the feature selection process, the model constructed with the remaining variables measured this value as 990.

The variables and the indicators that describe them are given in Part 2 of Table 4. Upon setting all variables to zero, our model yields the Intercept. "b" is the constant added to explain the beginning value of our line in the traditional " $y = mx + b$ " linear formula. We have our variables under the Intercept. The first informative column is the coefficient. It displays the magnitude and direction of each independent variable's change's impact on the dependent variable. To put it another way, it's the "m" in " $y = mx + b$ ". The dependent variable will be impacted by a unit change in the independent variable according to the coefficient's measure. There is an inverse relationship with the dependent variable if the coefficient is negative.

The standard error in a side column is an estimate of the coefficient's standard deviation; thus, it represents the amount of change in the coefficient as the data varies. The t-statistic measures the sensitivity of the coefficient. A low standard error when compared to a high coefficient results in a high t-statistic, indicating that the coefficient is highly significant. $P > jtj$ is one of the table's most important statistics. It is used to calculate the $p$-value, which is a measure of the likelihood that the coefficient was measured by chance using our model. As is known, a variable is considered significant when the $P$ value is <0.05. As seen in the table, since the $P$ value of the t-statistics for all variables is less than 0.05, we can say that all variables are significant for $CO_2$ emissions. [0.025 and 0.975] are both measurements of our coefficients within our data's 95% confidence interval.

## Comparison of regression models

In this stage of the analysis, a wide spectrum of machine learning-based regression models with different characteristics were applied to the experimental dataset. Initially, the data sets were divided into training and test subsets. This technique was used to evaluate the performance of machine learning models. While the training dataset was used to train the 18 machine learning-based regression models selected for our research, the test dataset was used to evaluate the models' performance. This investigation employs 18 different regression methods, six of which are based on deep learning techniques, six on ensemble learning, and six on conventional machine learning. For each algorithm, $R^2$, Adjusted $R^2$, RMSE, and runtime values were calculated to measure the performance metrics of each applied model. These values, ranked according to $R^2$, are provided in Table 5. The findings in the table compare the performance of the 18 regression models in terms of three learning types. Table 5 categorizes three learning models based on their average $R^2$ and separates the algorithms into these groups. The algorithms in each learning model are sorted in descending order based on their $R^2$ scores. According to the table, deep learning ($R^2 = 9,947.0$) is the most successful learning model, and the most effective algorithms in it are CNN, DNN, GRU, RNN, LSTM, and MLP, respectively, based on the $R^2$ score. The ensemble learning model ($R^2 = 9,873.8$), which is ranked second, is then listed, along with the algorithms that comprise it. The final row lists machine learning ($R^2 = 9,800.0$) along with the algorithms that fall under it.

In order to provide a deeper understanding of the results demonstrated in Table 5, we generated a visual representation of these results for every metric (see Fig. 1). The top ten algorithms based on $R^2$ were shown in Fig. 1A, followed by the top ten algorithms based

**Table 5  $R^2$, Adjusted $R^2$, RMSE, and runtime metrics of the regression models.**

| Learning type | Linearity | Model | $R^2$ | Adj. $R^2$ | RMSE | Runtime |
|---|---|---|---|---|---|---|
| | Nonlinear | CNN | 0.995672 | 0.995616 | 3.878894 | 30.873016 |
| | Nonlinear | DNN | 0.995548 | 0.995490 | 3.933941 | 20.109925 |
| Deep | Nonlinear | GRU | 0.995514 | 0.995511 | 3.949136 | 24.325117 |
| learning | Nonlinear | RNN | 0.994915 | 0.994912 | 4.204253 | 20.703259 |
| | Nonlinear | LSTM | 0.993801 | 0.993721 | 4.642037 | 156.227947 |
| | Nonlinear | MLP | 0.992927 | 0.992830 | 4.883358 | 14.745560 |
| | | Mean | 0.994730 | 0.994680 | 4.248603 | 42.830804 |
| | Nonlinear | XGB | 0.997909 | 0.997880 | 2.655423 | 0.100254 |
| | Nonlinear | RandomForest | 0.996638 | 0.996592 | 3.366763 | 1.834002 |
| Ensemble | Nonlinear | LGBM | 0.996331 | 0.996280 | 3.517346 | 0.080957 |
| learning | Nonlinear | HistGradientBoosting | 0.996331 | 0.996280 | 3.517382 | 0.534806 |
| | Nonlinear | GradientBoosting | 0.995221 | 0.995155 | 4.014151 | 0.513874 |
| | Nonlinear | AdaBoost | 0.941655 | 0.940853 | 14.025945 | 0.356160 |
| | | Mean | 0.987348 | 0.987173 | 5.182835 | 0.570009 |
| | Nonlinear | DecisionTree | 0.994136 | 0.994056 | 4.446513 | 0.024940 |
| | Linear | Ridge | 0.992803 | 0.992704 | 4.926218 | 0.010004 |
| Machine | Linear | LinearRegression | 0.992799 | 0.992700 | 4.927634 | 0.015624 |
| learning | Nonlinear | KNeighbors | 0.987786 | 0.987619 | 6.417273 | 0.041987 |
| | Linear | Lasso | 0.983382 | 0.983154 | 7.485444 | 0.071001 |
| | Nonlinear | SVR | 0.929334 | 0.928363 | 15.436015 | 3.264253 |
| | | Mean | 0.980040 | 0.979766 | 7.273183 | 0.571302 |

on RMSE in Fig. 1B, and the top ten algorithms based on runtime in Fig. 1C. According to Fig. 1A, the top five $R^2$ algorithms are XGB, RandomForest, LGBM, HistGradientBoosting, and CNN in that order. The primary reason why XGB delivers the best performance for estimating $CO_2$ emissions from fuel vehicles lies in its foundation as an enhanced version of the gradient boosting algorithm, which is capable of constructing complex models. This model incorporates mechanisms to prevent overfitting, such as L1 (Lasso) and L2 (ridge) regularization, allowing for more generalized and accurate predictions. Furthermore, XGBoost's ability to capture nonlinear relationships using decision trees provides a significant advantage in understanding the intricate and variable-relational structure of $CO_2$ emissions. The flexibility in hyper parameter tuning and the feature importance ranking capabilities enable the model to be optimized for higher accuracy predictions. These combined features make XGB the most effective model for $CO_2$ emission estimation of fuel vehicles.

Figure 1A also shows that the first four algorithms are ensemble learning-based. The three most effective algorithms that follow them, CNN, DNN, and GRU, are based on deep learning. As a result, ensemble models outperform traditional machine learning and deep learning models on this regression problem. The $R^2$ values of these algorithms are greater than 0.99, indicating that they are excellent predictors for this regression problem. The first four ensemble-based algorithms outperform the others in this experiment primarily because tree-based ensemble learning algorithms do not require preprocessing or feature

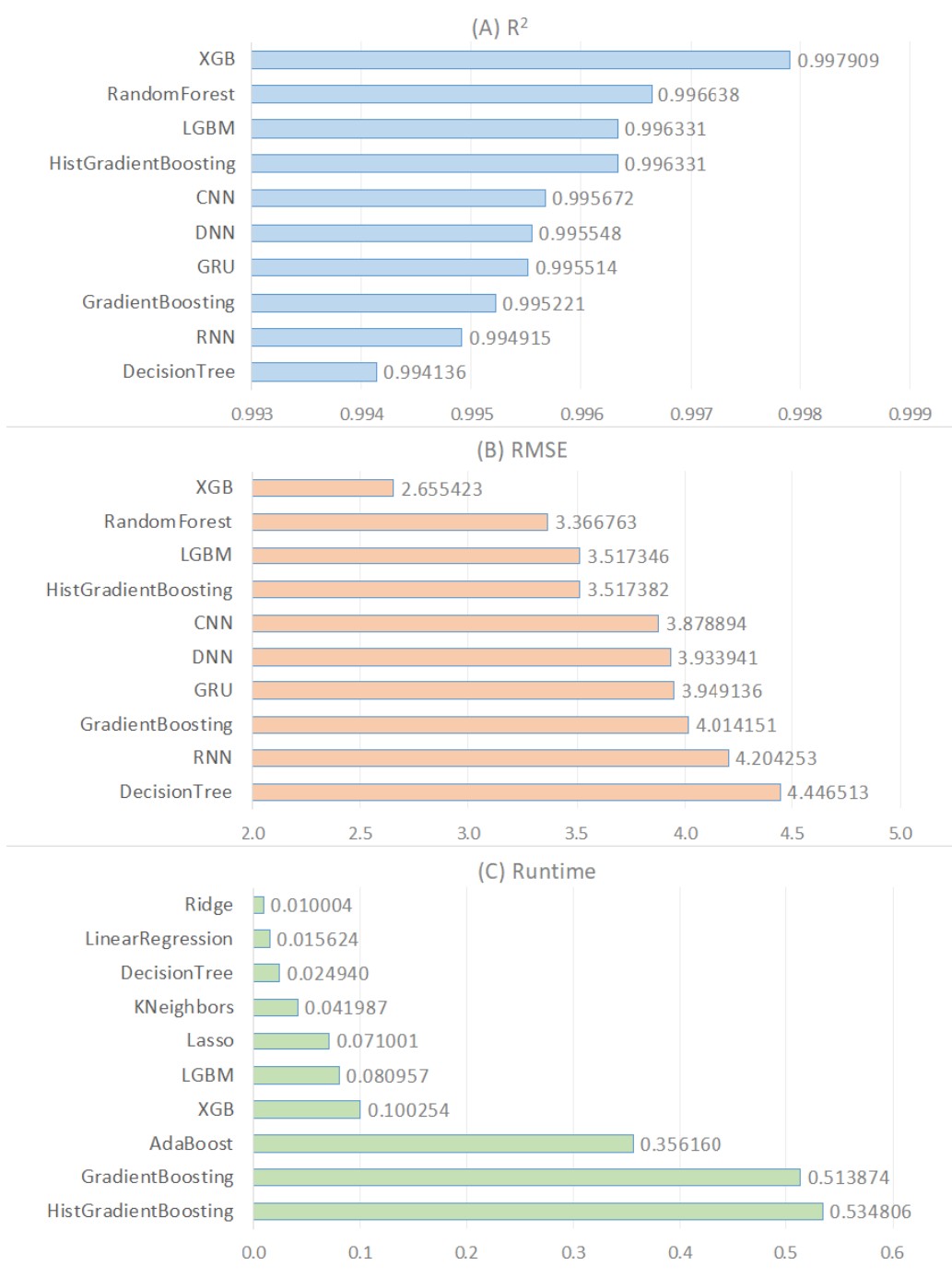

**Figure 1** Comparison of the top ten regression models for the R2, RMSE, and runtime metrics.

engineering processes like standardization or normalization. Other traditional machine learning algorithms, particularly linear-based statistical algorithms, require feature scaling to prevent features with a broad range from outperforming features with a narrow range. The second reason is that tree-based machine learning models improve the model through bagging and boosting techniques. Ensemble methods can make better predictions than a

single model because they combine predictions from multiple models. This leads to more accurate regression analysis. As a result, ensemble models produce significantly higher $R^2$ values for this regression problem. This indicates that the model-predicted $CO_2$ emissions values are very close to the actual values.

Figure 1B illustrates the performance of algorithms based on the RMSE metric, comparing the RMSE values of the top ten algorithms. RMSE, or Root Mean Square Error, is a commonly used measure to evaluate the performance of regression algorithms. RMSE is the square root of the average of the squares of the differences between the actual values and the predicted values of the model. A lower RMSE value indicates better performance of the model. According to Fig. 1B, the lowest RMSE value is obtained by the ensemble model "XGB" with a value of 2.655423. The ranking of other algorithms continues in the same order as Fig. 1A. This is because the higher the $R^2$ value, the smaller the RMSE value. Therefore, the algorithm XGB with the highest $R^2$ value also has the smallest RMSE value. Following XGB, the algorithms RandomForest and LGBM rank respectively.

Figure 1C demonstrates a comparison of the runtime (measured in seconds) of the various regression techniques used in this study. According to the figure, Ridge, LinearRegression, and DecisionTree are the regression algorithms with the lowest runtime, respectively. Considering Fig. 1C and Table 5, we observe that linear models based on traditional machine learning techniques reached results in significantly shorter times compared to ensemble learning and deep learning models in solving the regression problem in this analysis. In general, we conclude that traditional machine learning models provided solutions in a shorter time frame than ensemble learning and deep learning models in solving the regression problem in this analysis. In particular, deep learning models stand out as the algorithm with the longest running time (see Table 5). This is attributed to the fact that deep learning methodology involves more hidden layers and weightings compared to machine learning and ensemble learning. Additionally, our findings clearly demonstrate that linear models produce solutions much faster compared to nonlinear models (see Table 5).

Figure 2 presents a wide-ranging comparison of regression algorithms across three distinct learning paradigms: machine learning, deep learning, and ensemble learning, in terms of the metrics $R^2$, RMSE, and runtime. In Fig. 2A, deep learning emerges as the most effective learning approach based on the average $R^2$ value, while machine learning exhibits the weakest performance. Regarding the average RMSE value, ensemble learning algorithms are the most effective, followed by deep learning and machine learning (see Fig. 2B). Additionally, ensemble learning methods are identified as producing solutions much faster based on the average runtime measurement (see Fig. 2C). As a result, it is possible to conclude that regression models with deep learning are not particularly time efficient. Finally, as demonstrated in Table 5, nonlinear models outperform linear models in terms of $R^2$ and RMSE metrics while performing worse in terms of run time. In the context of ensemble learning, nonlinear models are the algorithms that perform the best

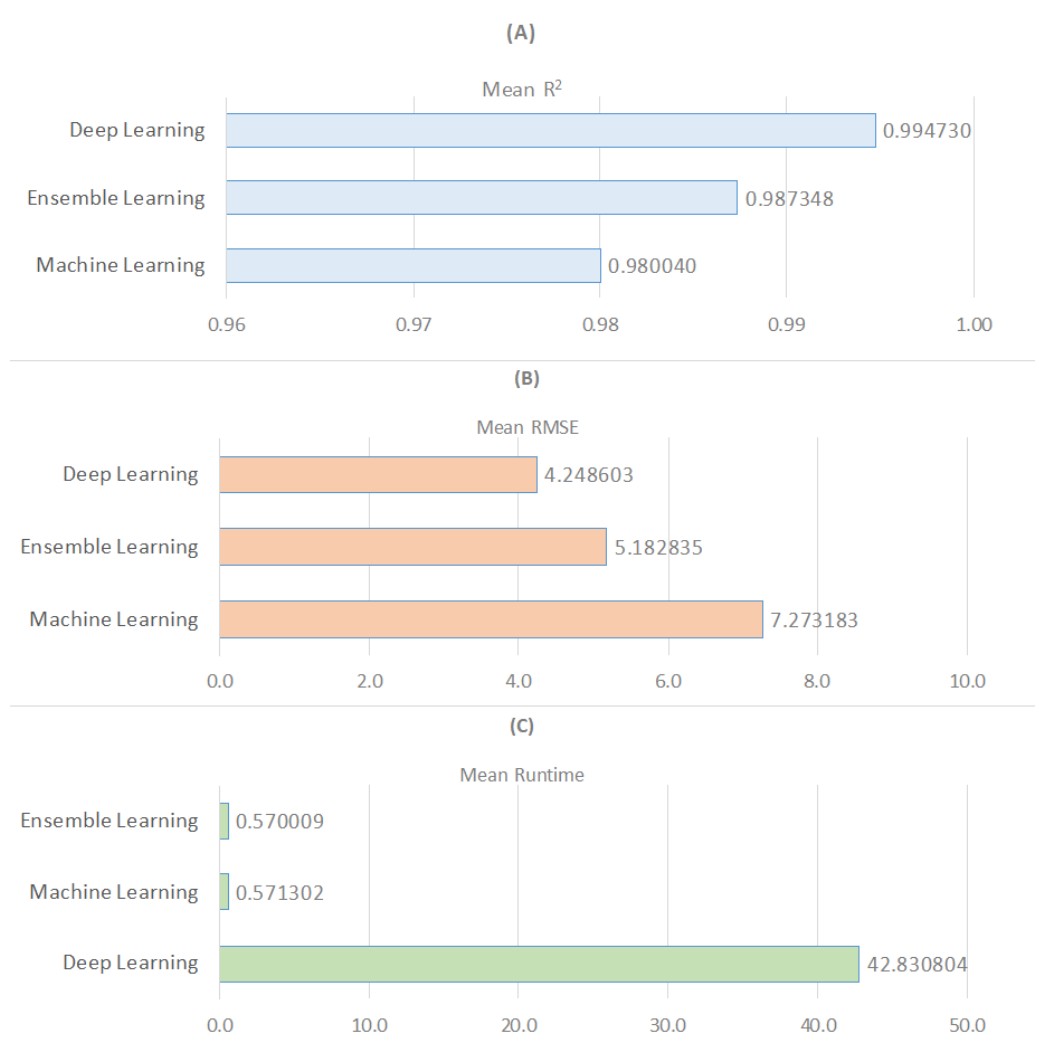

**Figure 2  Comparison of three learning models for R2, RMSE, and runtime metrics.**

predictions. This is followed by nonlinear models that fall under the category of deep learning.

## CONCLUSIONS

In this study, a comprehensive regression analysis comparing 18 different regression algorithms based on machine learning, ensemble learning, and deep learning paradigms for predicting $CO_2$ emissions of fuel-powered vehicles was conducted. In this analysis, the performance of each algorithm was evaluated using metrics such as $R^2$, Adjusted $R^2$, RMSE, and runtime. According to the results of the analysis, ensemble learning methods generally exhibited higher prediction accuracy and lower error rates. Particularly, ensemble algorithms such as XGB, RandomForest, and LGBM stood out with high $R^2$ values and low RMSE values, identified as the most effective methods for predicting $CO_2$ emissions. Some successful results were also obtained among deep learning models. In particular,

deep learning models with complex structures, such as CNN, DNN, and GRU, provided high $R^2$ values. However, it was observed that the training times of deep learning models were longer and required more computational resources.

Considering the average $R^2$ value, deep learning emerges as the most effective learning approach, while ensemble learning and machine learning exhibit lower performance. These findings show that regression models based on deep learning and ensemble learning are more effective methodologies for predicting vehicle $CO_2$ emissions and should be preferred in such analyses. Additionally, it was found that linear models based on traditional machine learning achieved results much faster in solving the regression problem in this analysis. On the other hand, it is important to choose the fitting method depending on the characteristics of the data set and the available resources. Limitations of the study include the dimension and characteristics of the dataset used, as well as the effects of the scaling and preprocessing methods employed. Future studies are needed to confirm and generalize these results with larger datasets and different modeling approaches. This study will guide future research and help us better understand the impact of regression analysis on predicting $CO_2$ emissions.

### Funding
The authors received no funding for this work.

### Competing Interests
The authors declare there are no competing interests.

### Author Contributions
- Fatih Gurcan conceived and designed the experiments, performed the experiments, analyzed the data, performed the computation work, prepared figures and/or tables, authored or reviewed drafts of the article, and approved the final draft.

### Data Availability
The source codes and dataset used in the experimental analysis are available in the Supplementary Files.

The third-part dataset collected from the Canada Government official website is available at Kaggle: https://www.kaggle.com/datasets/debajyotipodder/co2-emission-by-vehicles.

### Supplemental Information
Supplemental information for this article can be found online at http://dx.doi.org/10.7717/peerj-cs.2234#supplemental-information.

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
