# Peer review of "Forecasting CO2 emissions of fuel vehicles for an ecological world using ensemble learning, machine learning, and deep learning models"

_PeerJ Computer Science, doi:10.7717/peerj-cs.2234_

## Round 0.1 · original submission · Major Revisions

Please address the comments of the reviewers.

Reviewer 1 ·

Basic reporting

The author has addressed a relevant issue: CO2 emission vehicles' fuel consumption. this kind of research is needed for sustainable development. I have observed a few shortcomings in the manuscript, which are required to be incorporated before publication.
1> The flow of the introduction and the literature survey needs to be updated, along with the removal of grammatical errors. for example. line 108: environmentally friendly it should be "environment friendly".
2> In line 116, the research is required to be clearly mentioned, identified, and addressed.
3> A paragraph need to be added to state the organization of the work. for example: the rest of the work is organized as follows.................
4> The dataset utilized in the work is from 2014 to 2020, recent data can be used to make this study relevant as per the current statistics.

Experimental design

The authors have applied a list of machine-learning models. the ensemble learning-based model is applied and found good. A few details related to the ensembling of the model must be added.

The dataset description table can be added to present clear data statistics.

Validity of the findings

The performance of the applied model is good. the presentation of results is a little bit clumsy. the best-performing model should be clearly highlighted so that readers can use it for further studies.

Line number 484, Table A1, needs to be checked; it is not included in the manuscript.

Explain how the Adjusted R2 score is preferred over the R2 score in this work in terms of its significance.

·

Basic reporting

The author performed a comparative regression analysis using 26 different regression algorithms based on machine learning, ensemble learning, and deep learning paradigms to evaluate and predict CO2 emissions from fuel vehicles.
1. Clarity and Language Use. The language used in the paper is generally clear and professional. The sentences are well-constructed, and the terminology is appropriate for a scientific paper.
2. Literature reference and background. The paper provides a thorough review of existing literature, offering sufficient background and context in the field of the estimation of vehicle CO2 emissions.
3. Article Structure, Figures, and Tables. The overall structure of the article generally conforms to the acceptable format of 'standard sections.' However, I recommend revising the placement of figures and tables. Currently, all figures and tables are positioned after the conclusion, which makes it challenging to follow and understand their context within the related sections.
To improve clarity and facilitate easier comprehension, it would be better to insert figures and tables directly within the sections where they are discussed. This will allow readers to immediately reference the visual data and enhance their understanding of the text. Placing figures and tables in their corresponding sections will also improve the flow of the article and ensure that the data is presented in a coherent and logical manner.

Experimental design

The research question of the study should be clearly defined and relevant, addressing a meaningful knowledge gap. While the author identifies the contributions as preprocessing steps, feature selections, and a comparative analysis, these elements are standard practices in the field and do not present real novelty. These steps are common and often necessary in similar studies; thus, they cannot be considered significant contributions on their own.
Additionally, the title of the paper, "Forecasting CO2 emissions of fuel vehicles for an ecological world using ensemble learning, machine learning, and deep learning models," suggests a comprehensive exploration of various advanced deep learning models. However, among the 26 tested algorithms, only a basic MLP (Multilayer Perceptron) was included as a deep learning model. This could be misleading for readers, as more sophisticated deep learning architectures could have been tested to provide a broader and more in-depth analysis.
Moreover, the comparison of 26 different regression algorithms introduces ambiguity regarding the main goal of the study. It is unclear whether the primary focus is on forecasting CO2 emissions or on comparing regression algorithms. This lack of clarity makes it difficult to identify the main objective of the work.
Furthermore, there was no mention of the hyperparameters used for each algorithm. While this might be due to the high number of algorithms tested, it would have been more insightful to emphasize the best performing algorithms and discuss their modeling in greater depth. For example, detailing which hyperparameters were used, how the authors ensured there was no overfitting by plotting learning curves, and whether validation sets were used, would significantly enhance the study’s credibility and depth.
I recommend refining the research question to emphasize a specific, novel aspect of the study and clearly articulate how it fills an existing knowledge gap. Additionally, narrowing down the focus to a smaller, more relevant set of algorithms could help clarify the main goal and improve the study’s coherence and impact. Emphasizing and discussing the best performing algorithms in greater detail, would provide valuable insights and strengthen the study's contributions.

Validity of the findings

1- The study presents an extensive comparison of 26 different regression algorithms for forecasting CO2 emissions from fuel vehicles. While the comparative approach is thorough, several aspects raise concerns about the validity and practical significance of the findings.
The study's contributions i.e., preprocessing steps, feature selections, and comparative analysis are standard practices in the field and do not offer significant novelty. Without clear advancements or innovative methods, the findings may be considered derivative of existing work, reducing their value to the literature.
The title suggests a comprehensive analysis using ensemble learning, machine learning, and deep learning models. However, the study primarily relies on a basic MLP model for deep learning, which does not reflect the breadth of deep learning techniques available. This discrepancy could mislead readers and diminish the perceived validity of the findings.
The study attempts to achieve two main goals: forecasting CO2 emissions and comparing regression algorithms. This dual focus makes it challenging to discern the primary objective, potentially undermining the clarity and impact of the findings. Narrowing the focus to a smaller set of algorithms and providing a deeper analysis of their performance would enhance the study's coherence and relevance.
2- The data on which the conclusions are based has been provided and is publicly available, which is commendable. This dataset includes assessments of fuel consumption and measured CO2 emissions for a sample of 7,385 fuel-powered light commercial vehicles in Canada. It is accessible via the Canadian government’s official website and the Kaggle open data source.

---

## Round 0.2 · Minor Revisions

After a thorough review in the previous phase, there are still some minor considerations to further improve the presentation of the research conducted. Review the reviewer's comments to consider those that are appropriate or provide a response otherwise.

·

Basic reporting

The author performed a comparative regression analysis using 26 different regression algorithms based on machine learning, ensemble learning, and deep learning paradigms to evaluate and predict CO2 emissions from fuel vehicles.

1. Clarity and Language Use. The language used in the paper is generally clear and professional. The sentences are well-constructed, and the terminology is appropriate for a scientific paper.
One thing that can be changed is that the author refers to the algorithms using their class names from the libraries, such as CNNRegressor, DNNRegressor, and GRURegressor. While this is understandable from a coding and implementation perspective, it would be clearer for the reader if the standard algorithm names were used, such as Convolutional Neural Networks (CNN), Deep Neural Networks (DNN), and Gated Recurrent Units (GRU).

Switching to the standard algorithm names will improve the clarity and readability of the paper and would :

Make the paper more accessible to readers who are familiar with these algorithms but may not recognize the specific class names used in the code.
Avoid potential confusion and ensure consistency with the broader literature.

2. Literature reference and background. The paper provides a thorough review of existing literature, offering sufficient background and context in the field of the estimation of vehicle CO2 emissions.

3. Article Structure, Figures, and Tables. The overall structure of the article generally conforms to the acceptable format of 'standard sections.'

Experimental design

The author has made significant improvements to the paper based on the initial review:

Reduced Number of Compared Algorithms: The number of compared algorithms has been lowered, which helps to clarify the main goal of the study. This change makes it easier to focus on the primary objective of forecasting CO2 emissions.

Inclusion of More Deep Learning Models: Additional deep learning models have been included, making the study consistent with the title, "Forecasting CO2 emissions of fuel vehicles for an ecological world using ensemble learning, machine learning, and deep learning models." This provides a broader and more comprehensive analysis.

Clearer Research Question and Contributions: The research question has been better articulated, and the contributions have been more clearly defined. This refinement helps in understanding how the study aims to fill an existing knowledge gap.

However, despite these improvements, I still feel that the study lacks a real novelty in the field, To make a more substantial contribution, the study could explore more innovative approaches or novel insights.

Validity of the findings

The study has undergone several important revisions that have made the research easier to follow and more consistent with the title:

Reduced Number of Compared Algorithms: The number of compared regression algorithms has been lowered from 26 to 18. This reduction helps to streamline the analysis and makes the study more focused and coherent.

Addition of More Deep Learning Algorithms: The inclusion of additional deep learning models enhances the study's comprehensiveness. This makes the title, "Forecasting CO2 emissions of fuel vehicles for an ecological world using ensemble learning, machine learning, and deep learning models," more accurate and reflective of the study's content.

The data on which the conclusions are based has been provided and is publicly available, which is commendable. This dataset includes assessments of fuel consumption and measured CO2 emissions for a sample of 7,385 fuel-powered light commercial vehicles in Canada. It is accessible via the Canadian government’s official website and the Kaggle open data source.

---

## Round 0.3 · accepted · Accept

Congratulations on the acceptance of your manuscript, thank you for have addressed all of the reviewers' comments.